# Increased risk of cardiac arrhythmia in Hailey-Hailey disease patients

**William Jebril[1,2], Philip Curman[1,2,3], Daniel C. Andersson[4,5], Henrik Larsson[6], Etty Bachar-Wikstrom[1], Martin Cederlöf[3,6], Jakob D. Wikstrom[1,2]***

**1** Dermatology and Venereology Division, Department of Medicine (Solna), Karolinska Institutet, Stockholm, Sweden, **2** Dermato-Venereology Clinic, Karolinska University Hospital, Stockholm, Sweden, **3** Department of Medical Epidemiology and Biostatistics (Solna), Karolinska Institutet, Stockholm, Sweden, **4** Department of Physiology and Pharmacology, Karolinska Institutet, Stockholm, Sweden, **5** Cardiology Unit, Heart, Vascular and Neurology Theme, Karolinska University Hospital, Stockholm, Sweden, **6** School of Medical Sciences, Faculty of Medicine and Health, Örebro University, Örebro, Sweden

* jakob.wikstrom@ki.se

## Abstract

### Background

Hailey-Hailey disease (HHD) is a rare autosomal dominant skin disease caused by mutations in the *ATP2C1* gene, which encodes the secretory $Ca^{2+}/Mn^{2+}$-ATPase (SPCA1) pump in the Golgi apparatus. Although *ATP2C1* is ubiquitously expressed in the body, possible extracutaneous manifestations of HHD are unknown. However, dysfunction of the Golgi apparatus not specifically coupled to *ATP2C1* has been associated with heart disease.

### Objective

To investigate the association between HHD and common heart disease in a Swedish, population-based cohort.

### Methods

We conducted a population-based cohort study based on a linkage of Swedish nationwide registers to investigate the relationship between HHD and heart disease. We have been granted ethical approval from the Swedish Ethical Review Authority to conduct this study. The patients in this manuscript have given written informed consent to the publication of their case details. A total of 342 individuals with an ICD-10 diagnosis of HHD (Q82.8E) were identified and matched with randomly selected comparison individuals without HHD on a 1:100 ratio. Furthermore, in a separate clinical cohort we matched 23 HHD patients for age, sex, and BMI with control subjects to examine electrocardiogram parameters, electrolytes, and cardiovascular biomarkers.

### Results

Compared with individuals without HHD, individuals with HHD had an excess risk of arrhythmia (RR 1.4, CI 1.0–2.0), whereas no increased risks of myocardial infarction (RR 1.1, CI

**Data Availability Statement:** All relevant data are within the manuscript and its Supporting Information files.

**Funding:** This work was supported by grants from Hudfonden, Swedish Science Council, Swedish Society for Medical Research, Leo foundation, ALF medicin Stockholm, Jeanssons stiftelse, Wallenberg foundation and Tore Nilssons Stiftelse. All grants were to Jakob D Wikstrom. The funders had no role in study design, data collection and analysis, decision to publish, or preparation of the manuscript.

**Competing interests:** The authors have declared that no competing interests exist.

0.6–1.7) or heart failure (RR 1.0, CI 0.6–1.6; Table 1) were found. We found no difference in ECG parameters, cardiovascular biomarkers, and electrolytes in the clinical subset.

## Conclusion

This study reveals that HHD is associated with an increased risk of arrhythmia and represents the first data of any extracutaneous comorbidity in HHD. Thus, HHD may be a systemic disease. Our findings also shed light on the importance of the Golgi apparatus' $Ca^{2+}/Mn^{2+}$ homeostasis in common heart disease.

## Introduction

Hailey-Hailey disease (HHD) is a rare skin disease with an autosomal dominant inheritance pattern and complete penetrance. Symptoms primarily arise in early adulthood, with painful

**Table 1. Risk of myocardial infarction, heart failure, and arrhythmia in individuals with HHD.** A total of 342 individuals with an ICD-10 diagnosis of HHD (Q82.8E) were identified and matched with comparison individuals without HHD on a 1:100 ratio, randomly selected from the general Swedish population (all individuals with HHD diagnosis since the start of the register included). Successful matching was performed for birth year, sex, and county of residence at the time of the first HHD diagnosis of the individual. This matching scheme is referred to as incidence density sampling. Conditional logistic regression analyses were performed for the associations between HHD and the major ICD groups of heart diseases: myocardial infarction [I21], heart failure [I42, I50], and arrhythmias. SAS 9.3 software (SAS Institute, Cary, NC) was used for statistical analyses. The results were expressed as odds ratios and corresponding 95% confidence intervals. As a result of the incidence density sampling, odds ratios can be interpreted as risk ratios (RR). The table also shows the mean age and standard deviation (SD) at first arrhythmia diagnosis among individuals with HHD and comparison individuals. *Diagnoses included in arrhythmias: paroxysmal tachycardia [I47], atrial fibrillation and flutter [I48], and other cardiac arrhythmias [I49]. Within the group called other cardiac arrhythmias, there are ten arrhythmias, including ventricular fibrillation [I49.01], ventricular flutter [I49.02], atrial premature depolarization [I49.1], junctional premature polarization [I49.2], ventricular premature polarization [I49.3], unspecified premature polarization [I49.40], other premature polarization [I49.49], sick sinus syndrome [I49.5], other specified cardiac arrhythmias [I49.8] and cardiac arrhythmia, unspecified [I49.9].

| Variable | Sample size | | Mean ± standard deviation | | Unpaired t-Test |
|---|---|---|---|---|---|
| | HHD | Control | HHD | Control | P-value |
| **Age** | 23 | 23 | 53.6 ± 10 (33–75) | 51.5 ± 10.9 (33–70) | 0.51 |
| **Sex** | 23 (7M, 16F)** | 23 (7M, 16F)** | | | |
| **ECG parameters** | | | | | |
| Heart rate (per minute) | 23 | 23 | 63.30 ± 11.060 (45–85) | 63.21 ± 10.023 (47–84) | 0.977 |
| PQ—interval (ms) | 23 | 23 | 157.66 ± 20.482 (116–194) | 155 ± 24.462 (108–210) | 0.695 |
| QRS—duration (ms) | 23 | 23 | 92 ± 10.531 (78–112) | 92.35 ± 9.810 (74–114) | 0.906 |
| QT—interval (ms) | 23 | 23 | 411.57 ± 30.960 (348–498) | 413.30 ± 28.855 (354–470) | 0.845 |
| QTc -interval (ms) Bazzet formula | 23 | 23 | 417.74 ± 20.633 (379–466) | 420.61 ± 24.062 (368–455) | 0.666 |
| QTc-interval (ms) Fridericia formula | 23 | 23 | 416.10 ± 31.52 (388–456) | 418.42 ± 20.86 (378–448) | 0.301 |
| QTc–interval (ms) Framingham formula | 23 | 23 | 414.99 ± 38.99 (385–451) | 417.65 ± 21.65 (372–448) | 0.649 |
| QTc–interval (ms) Hodges formula | 23 | 23 | 414.35 ± 36.74 (387–474) | 418.24 ± 19.57 (382–456) | 0.333 |
| **Blood biomarkers and electrolytes** | | | | | |
| NT-proBNP (ng/L) | 23 | 23 | 66.43 ± 55.717 (6–230) | 94.09 ± 191.369 (11–954) | 0.509 |
| Troponin T (ng/L) | 23 | 23 | 7.83 ± 10.360 (5–55) | 5.26 ± 0.619 (5–7) | 0.845 |
| Sodium | 23 | 23 | 140.87 ± 1.817 (138–145) | 140.43 ± 1.805 (136–144) | 0.420 |
| Potassium | 23 | 23 | 4.09 ± 0.251 (3.7–4.6) | 4.02 ± 0.332 (3.4–4.8) | 0.399 |
| Calcium | 23 | 23 | 2.39 ± 0.090 (2.22–2.58) | 2.34 ± 0.071 (2.22–2.49) | 0.037* |

recurring blisters and erosions. The disease has a predilection for skin folds, such as the axillae and groin area, and generalized skin symptoms are uncommon. Typically, a ubiquitously expressed mutation in the *ATP2C1* gene that encodes the secretory $Ca^{2+}/Mn^{2+}$-ATPase (SPCA1) pump in the Golgi apparatus (GA) is the underlying cause. This is similar to Darier disease, which is caused by mutations in the *ATP2A2* gene that encodes the sarcoendoplasmic reticulum $Ca^{2+}$ pump (SERCA2). While Darier disease is increasingly recognized as a systemic condition due to emerging evidence of extracutaneous involvement [1–4], no such evidence exists for HHD to date. It is, however, plausible, due to the ubiquitous expression of the disease-causing gene. Heart disease in particular is a likely comorbidity because of the importance of GA in heart excitation-contraction coupling [5]. Herein, we examine for the first time the association between HHD and common heart disease by utilizing a population-based cohort based on Swedish national register data, as well as a clinical patient cohort.

## Materials and methods

We have been granted ethical approval from the Swedish Ethical Review Authority to conduct this study. The patients in this manuscript have given written informed consent to the publication of their case details.

First, we conducted a population-based cohort study based on linkage between Swedish nationwide registers to investigate the relationship between HHD and cardiac abnormalities. The Total Population Register [6] holds demographic information of all Swedish inhabitants since 1968; The National Patient Register [7] includes inpatient diagnoses assigned by the treating physicians according to the International Classification of Diseases, since 1973 and outpatient diagnoses since 2001 [8]. 342 individuals with an ICD-10 diagnosis of HHD (Q82.8E) were identified and matched with comparison individuals without HHD on a 1:100 ratio. Comparison individuals were randomly selected from the general Swedish population. Successful matching was performed for birth year, sex, and county of residence at the time of the first HHD diagnosis of the index persons. This matching scheme is referred to as incidence density sampling. Conditional logistic regression analyses were performed for the associations between HHD and the major ICD groups of heart diseases (myocardial infarction [I21], heart failure [I42, I50], and arrhythmias [I47-49]) that could be delineated with sufficient statistical power. The results were expressed as odds ratios with corresponding 95% confidence intervals using the SAS 9.3 software (SAS Institute, Cary, NC).

Secondly, from 2017-10-01 to 2017-11-01, we recruited 23 HHD (7 males and 16 females) patients from the dermatology department at the Karolinska University Hospital (Stockholm, Sweden). Inclusion criteria were phenotype-positive patients with a histopathological verified HHD and a family history of HHD. A family history of HHD was defined as the presence of a first- or second-degree relative with the disease verified at a dermatology clinic or hospital. Healthy controls were similarly recruited through advertisements at the dermatology clinic and on the Karolinska Institutet (Stockholm, Sweden) website. Patients underwent a thorough medical history and physical skin examination. We subsequently matched the HHD patients for age, sex, and BMI with healthy controls on a 1:1 ratio to investigate 12-lead electrocardiogram parameters, blood biomarkers (NT-proBNP, troponin T, sodium, potassium and calcium), and electrolytes to perform a clinical study to supplement our population-based study.

## Results

In the population-based study, individuals with HHD showed an excess risk of arrhythmia diagnoses (RR 1.4, CI 1.0–2.0), whereas no statistically significant elevated risks could be

**Table 2. ECG parameters, blood biomarkers, and electrolytes from 23 HHD patients and matched healthy control subjects.** Inclusion criteria were a diagnosis of HHD set by a dermatologist based on typical clinical appearance, histopathology, and family history. Exclusion criteria were age <18 years, current pregnancy, active substance abuse, and acute illness in the past 4 weeks. The control group was matched for age, sex, and BMI.

| | Myocardial infarction | | Heart failure | | Arrhythmia* | | |
|---|---|---|---|---|---|---|---|
| | N (%) | RR (CI) | N (%) | RR (CI) | N (%) | RR (CI) | Mean age in years (CI) at first diagnosis |
| Individuals with HHD, N = 342 | 20 (5.9) | 1.1 (0.6–1.8) | 20 (5.9) | 1.1 (0.6–1.7) | 40 (11.7) | 1.4 (1.0–2.0) | 67.7 (63.5:72.0) |
| Comparison individuals without HHD, N = 34,200 | 1,917 (5.6) | | 1,917 (5.6) | | 3,047 (8.9) | | 71.6.0 (71.5;71.6) |

*After multiple comparisons with Bonferroni post hoc correction, none of the p-values are significant (p<0.005).

** M = males, F = females.

confirmed for myocardial infarction (RR 1.1, CI 0.6–1.7), or heart failure (RR 1.0, CI 0.6–1.6; Table 1).

None of the patients in the clinical study suffered from cardiac diseases or arrhythmia, nor did they take medications or underlying comorbidities that could trigger cardiac disturbances. We found no statistically significant differences between the blood chemistry and EKG variables (Table 2). Total $Ca^{2+}$ was slightly higher in the HHD group, although within the normal physiological range, and after multiple comparison corrections, this finding was insignificant.

## Discussion

Our population-based study reveal that HHD is associated with an increased risk of arrhythmia, and these findings represent the first data of any extracutaneous comorbidity in HHD.

### Skin disease and heart comorbidities

Several skin diseases are linked with heart comorbidities. For example, the inflammation in common psoriasis and hidradenitis suppurativa is thought to cause a higher prevalence of the cardiovascular disease. Numerous rare genetic skin disorders also have a skin-heart connection. H syndrome, caused by autosomal recessive mutations in the *SLC29A3* gene that encodes the human equilibrative nucleoside transporter 3 (hENT3), a protein found in endosomes, lysosomes, and mitochondria, is clinically characterized by cutaneous hyperpigmentation, hypertrichosis, hepatosplenomegaly, and hypogonadism as well as heart anomalies [9]. Tuberous sclerosis complex (TSC), caused by autosomal dominant mutations in either *TSC1* or *TSC2* that causes a plethora of cellular dysfunctions due to mTOR inhibition, has heart symptoms in the form of rhabdomyomas in addition to other organ involvements such as the skin and brain [10]. Darier disease, which is pathophysiologically very similar to HHD as it is caused by mutations in *ATP2A2* that encodes the SERCA2, thus upstream in the cells' secretory pathway, is associated with heart failure [1] as well as diabetes [3], cognitive impairment [2] and several psychiatric disorders [11].

There are several examples of monogenic syndromes with congenital heart disease that also show skin pathologies such as loose skin (Marfan Syndrome), excess nuchal skin (Noonan syndrome), skin tags (Duane-radial ray syndrome), and dark skin (Costello syndrome) [12]. Furthermore, arrhythmogenic right ventricular cardiomyopathy (ARVC), usually caused by mutations in *PKP2* that encode for the desmosome-related protein plakophilin 2, causes loss of cell-to-cell adhesion in both skin and heart [13]. Desmosomal dysfunction is, in fact, considered the underlying cause in several conditions with a heart-skin connection [14, 15]. Our finding is that HHD, histologically characterized by acantholysis due to loss of intercellular

connections, is associated with arrhythmia and thus fits with previous data from other diseases. Apart from a single case of suspected liver involvement [16], extracutaneous manifestations are unknown in HHD.

## Golgi apparatus and heart disease

The Golgi apparatus (GA) is a complex organelle in the secretory pathway and is divided into compartments. The cis-Golgi receives lipids and proteins from the endoplasmic reticulum, while the trans-Golgi packages and transports modified proteins. The GA is responsible for lipidation and glycosylation and also plays a vital role in cell signaling and regulation of protein activity [5]. Furthermore, the GA forms lysosomes which degrade and recycle molecules delivered by endocytosis, phagocytosis, and autophagy. These Golgi functions are crucial for most cells, including cardiomyocytes. Lysosome dysfunction has been implicated in Danon disease, a rare x-linked lysosomal disease with cardiomyopathy, skeletal myopathy, and cognitive dysfunction [17]. Since the GA also is responsible for directing newly synthesized cardiac ion channel proteins to their appropriate sarcolemma-bound locations [18], and dysfunction of the GA has been associated with atrial fibrillation when sections of the left atrial free wall were analyzed [19], it is likely that perturbations to the GA could play a role in excitation–contraction coupling and the cardiac action potential which may explain the increased frequency of arrhythmia in HHD. Although further mechanistic research is needed, our study indicates an important role for the SPCA1 pump in the heart. Extracutaneous manifestations are unknown in HHD, and the potential systemic effects of *ATP2C1* mutations have not been studied. This paper provides new knowledge on the relationship between the GA and HHD and their cardiac manifestations.

## SPCA1 function and cardiac health

To the best of our knowledge, there are no available studies indicating that SPCA1 dysfunction leads to heart disease in humans. In terms of preclinical studies, there is only one study that examines heart development in homozygote SPCA1 knock-out mice in which embryos died in uteri however with no apparent cardiac pathology [20]. However, it is likely that SPCA1 plays an important role in the heart because of its' role in intracellular calcium homeostasis and that SPCA1 activity was reported high in the heart [21]. Future studies should examine cardiac function in aged SPCA1 heterozygote knock-out or perhaps better knock-in animals HHD like mutations. Regarding general aging, there is no direct evidence that SPCA1 plays a role, however it was shown that in ultraviolet light exposed keratinocytes, low levels of SPCA1 increases cytosolic calcium as well as reactive oxygen levels, which may contribute to cellular aging [22].

## Strengths and limitations

The main strength of this study is that we used a large comprehensive patient registry with physician assigned diagnoses that enabled us to link a rare diagnosis, HHD, to common heart disease diagnoses. A limitation of the register-based study is that the specific diagnosis codes have not been formally validated, however, a validation study has shown that diagnoses of chronic disorders are generally valid [7]. Also, a specialist dermatologist almost exclusively assigns diagnoses of HHD after a thorough work-up, including the clinical presentation, family history, histopathological findings, and genetic testing. Another possible limitation is the potential influence of unknown confounding factors not accounted for in the population-based data. Further, our clinical cohort of HHD patients is the world's largest to our best

knowledge, yet the cohort's limited size and thus power might have led to statistical type I error.

## Conclusions

In the population-based cohort study, we found an excess risk of arrhythmia diagnosis among HHD patients. These findings expand our understanding of HHD, as it potentially may be a systemic condition not just confined to the skin. Our study also sheds light on the importance of the Golgi apparatus in cardiac physiology. HHD patients may need to be screened for heart disease in clinical practice; however, larger cohort clinical studies are needed to achieve sufficient power.

## Supporting information

**S1 Table. Risk of myocardial infarction, heart failure, and arrhythmia in individuals with HHD.** A total of 342 individuals with an ICD-10 diagnosis of HHD (Q82.8E) were identified and matched with comparison individuals without HHD on a 1:100 ratio, randomly selected from the general Swedish population (all individuals with HHD diagnosis since the start of the register included). Successful matching was performed for birth year, sex, and county of residence at the time of the first HHD diagnosis of the individual. This matching scheme is referred to as incidence density sampling. Conditional logistic regression analyses were performed for the associations between HHD and the major ICD groups of heart diseases: myocardial infarction [I21], heart failure [I42, I50], and arrhythmias. SAS 9.3 software (SAS Institute, Cary, NC) was used for statistical analyses. The results were expressed as odds ratios and corresponding 95% confidence intervals. As a result of the incidence density sampling, odds ratios can be interpreted as risk ratios (RR). The table also shows the mean age and standard deviation (SD) at first arrhythmia diagnosis among individuals with HHD and comparison individuals. *Diagnoses included in arrhythmias: paroxysmal tachycardia [I47], atrial fibrillation and flutter [I48], and other cardiac arrhythmias [I49]. Within the group called other cardiac arrhythmias, there are ten arrhythmias, including ventricular fibrillation [I49.01], ventricular flutter [I49.02], atrial premature depolarization [I49.1], junctional premature polarization [I49.2], ventricular premature polarization [I49.3], unspecified premature polarization [I49.40], other premature polarization [I49.49], sick sinus syndrome [I49.5], other specified cardiac arrhythmias [I49.8] and cardiac arrhythmia, unspecified [I49.9].
(DOCX)

**S2 Table. ECG parameters, blood biomarkers, and electrolytes from 23 HHD patients and matched healthy control subjects.** Inclusion criteria were a diagnosis of HHD set by a dermatologist based on typical clinical appearance, histopathology, and family history. Exclusion criteria were age <18 years, current pregnancy, active substance abuse, and acute illness in the past 4 weeks. The control group was matched for age, sex, and BMI. *After multiple comparisons with Bonferroni post hoc correction, none of the p-values are significant (p<0.005). ** M = males, F = females.
(DOCX)

## Author Contributions

**Conceptualization:** Philip Curman, Daniel C. Andersson, Henrik Larsson, Etty Bachar-Wikstrom, Martin Cederlöf, Jakob D. Wikstrom.

**Data curation:** William Jebril, Philip Curman, Martin Cederlöf.

**Formal analysis:** William Jebril, Philip Curman, Martin Cederlöf.

**Funding acquisition:** Jakob D. Wikstrom.

**Investigation:** William Jebril, Philip Curman, Martin Cederlöf.

**Methodology:** William Jebril, Martin Cederlöf.

**Project administration:** Etty Bachar-Wikstrom, Jakob D. Wikstrom.

**Supervision:** Daniel C. Andersson, Henrik Larsson, Martin Cederlöf, Jakob D. Wikstrom.

**Writing – original draft:** William Jebril, Philip Curman, Daniel C. Andersson, Henrik Larsson, Etty Bachar-Wikstrom, Martin Cederlöf, Jakob D. Wikstrom.

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
