## [Decision Letter · Decision Letter 0]

7 Feb 2024

PONE-D-23-25220Increased risk of cardiac arrhythmia in Hailey-Hailey disease patientsPLOS ONE

Dear Dr. Wikstrom,

Thank you for submitting your manuscript to PLOS ONE. After careful consideration, we feel that it has merit but does not fully meet PLOS ONE’s publication criteria as it currently stands. Therefore, we invite you to submit a revised version of the manuscript that addresses the points raised during the review process. The review process has revealed need for clarification and I would ask you to respond to the comments in detail, especially as one reviewer has raised significant reservation regarding the methodology.

We look forward to receiving your revised manuscript.

Kind regards,

Albert Rübben, Ass. Prof., M.D., Ph.D.

Academic Editor

PLOS ONE

Journal Requirements:

4. In the online submission form, you indicated that the datasets generated during and/or analysed during the current study are available from the corresponding author on reasonable request.

Reviewer's Responses to Questions

**Comments to the Author**

1. Is the manuscript technically sound, and do the data support the conclusions?

Reviewer #1: No

Reviewer #2: Yes

2. Has the statistical analysis been performed appropriately and rigorously? 

Reviewer #1: Yes

Reviewer #2: Yes

3. Have the authors made all data underlying the findings in their manuscript fully available?

Reviewer #1: Yes

Reviewer #2: Yes

4. Is the manuscript presented in an intelligible fashion and written in standard English?

Reviewer #1: Yes

Reviewer #2: Yes

5. Review Comments to the Author

Reviewer #1: Major issues

1. This reviewer does not think that the data presented in this manuscript explain and justify the conclusions made by the authors, i.e. patients with HHD would be at a significantly higher risk for cardiac arrhythmias. Although it is an interesting assumption, however, a more careful selection of patients as well as mechanistic studies should be performed to establish such a link. It is not mentioned what medications were taken by patients in any of the groups, a number of cardiovascular and non-cardiovascular drugs can cause arrhythmias. It is not mentioned what other pathologies were present in any of the two groups, that could be associated with arhythmias. It is not mentioned how many male/female patients were included in the different groups.

2. A large number of different types of cardiac arrhythmias were reported in the population-based cohort in the HHD group, including supraventricular arrhythmias, even atrial fibrillation, then junctional arrhythmias and ventricular arrhythmias, including ventricular fibrillation. In HHD patients, how many had heart failure, previous myocardial infarction, any condition that could lead to arrhythmia development (hyperthyreosis etc)?

3. The conclusions are also not supported by the "separate clinical cohort" (?), with absolutely no differences in ECG parameters, no arrhythmia development etc.

Minor issues

1. It is recommended to state the method how the frequency-corrected QT interval was calculated. The literature suggests that the Basett's and Fridericia corrections tend to over- or undercorrect at heart rates markedly deviating from textbook average basal heart rates. Use of the Hodges, Framingham etc corrections are recommended.

2. I do not think that 2.39 vs 2.34 (probably mmol/L, not shown in manuscript) in calcium plasma levels is a meaningful clinical difference that would explain any conclusions, even if this is a mathematically significant difference.

Reviewer #2: In the manuscript, the authors present the results of the study, in which they investigate the relationship between Hailey-Hailey disease and associated cardiac arrythmia in a Swedish population-based cohort.

Results are clinically relevant and reviles new knowledge on the cardiac manifestations in Hailey-Hailey disease.

The manuscript is well prepared.

Comment:

Blood biomarkers are mentioned in line 114. I would suggest that the authors explain which biomarkers were investigated.

6. PLOS authors have the option to publish the peer review history of their article (what does this mean?). If published, this will include your full peer review and any attached files.

Reviewer #1: No

Reviewer #2: No

---

## [Author Response · Author response to Decision Letter 0]

22 Jul 2024

Reviewer #1: Major issues

1. This reviewer does not think that the data presented in this manuscript explain and justify the conclusions made by the authors, i.e. patients with HHD would be at a significantly higher risk for cardiac arrhythmias. Although it is an interesting assumption, however, a more careful selection of patients as well as mechanistic studies should be performed to establish such a link. It is not mentioned what medications were taken by patients in any of the groups, a number of cardiovascular and non-cardiovascular drugs can cause arrhythmias. It is not mentioned what other pathologies were present in any of the two groups, that could be associated with arhythmias. It is not mentioned how many male/female patients were included in the different groups.

Answer: First, we would like thank the reviewer for investing the time to review our manuscript. We agree that more studies to ascertain the link to arrythmias are needed and therefore we have softened this conclusion (line 205, pages 8). As this is a clinical and population-based registry study we cannot perform mechanistic experiments to study the underlying molecular pathology. Also, given the nature of the register-based data it is unfortunately unfeasible to correct for the vast number of possible drugs that can give rise to cardiac arrhythmias, even though we agree that this would be interesting to do. 

The male/female ratio is now included (lines 106 and 107, page 5). None of the patients in the clinical study suffered from cardiac diseases or arrhythmia nor did they take medications or underlying comorbidities that could trigger cardiac disturbances (page 5, line 123-126).

2. A large number of different types of cardiac arrhythmias were reported in the population-based cohort in the HHD group, including supraventricular arrhythmias, even atrial fibrillation, then junctional arrhythmias and ventricular arrhythmias, including ventricular fibrillation. In HHD patients, how many had heart failure, previous myocardial infarction, any condition that could lead to arrhythmia development (hyperthyreosis etc)?

Answer: Thank you for pointing to possible confounding factors that might contribute to the findings in the population-based cohort. Given the nature of the population-based data we are unable to correct for these underlying conditions at this time. We can, however, reason that the same factors that give rise to cardiac arrhythmias also could give rise to heart failure and other cardiac-related disorders, why the association observed still holds true. We have added a sentence on this in the limitations section on line 201-202. 

3. The conclusions are also not supported by the "separate clinical cohort" (?), with absolutely no differences in ECG parameters, no arrhythmia development etc.

Answer: HHD is a very rare disease with an incidence of 1:50,000. The cohort of 23 patients examined belong to the largest cohorts studied worldwide and is really the best we could achieve in a single center study. We have added a statement that studies on larger cohorts are needed to achieve sufficient power (line 205, page 8).

Minor issues

1. It is recommended to state the method how the frequency-corrected QT interval was calculated. The literature suggests that the Basett's and Fridericia corrections tend to over- or undercorrect at heart rates markedly deviating from textbook average basal heart rates. Use of the Hodges, Framingham etc corrections are recommended.

Answer: Thank you for this useful comment. We have included calculations using the Bazzet formula, Fridericia formula, Framingham formula, and Hodges formula in Table 2.

2. I do not think that 2.39 vs 2.34 (probably mmol/L, not shown in manuscript) in calcium plasma levels is a meaningful clinical difference that would explain any conclusions, even if this is a mathematically significant difference.

Answer: In the result section we state that “Total Ca2+ was slightly higher in the HHD group, although within the normal physiological range and after multiple comparison correction this finding was insignificant.”

Reviewer #2: In the manuscript, the authors present the results of the study, in which they investigate the relationship between Hailey-Hailey disease and associated cardiac arrythmia in a Swedish population-based cohort.

Results are clinically relevant and reviles new knowledge on the cardiac manifestations in Hailey-Hailey disease.

The manuscript is well prepared.

Answer: We thank the reviewer for the positive outlook on the manuscript.

Comment:

Blood biomarkers are mentioned in line 114. I would suggest that the authors explain which biomarkers were investigated.

Answer: The blood biomarkers are shown in Table 2.

---

## [Decision Letter · Decision Letter 1]

13 Aug 2024

Increased risk of cardiac arrhythmia in Hailey-Hailey disease patients

PONE-D-23-25220R1

Dear Dr. Wikstrom,

We’re pleased to inform you that your manuscript has been judged scientifically suitable for publication and will be formally accepted for publication once it meets all outstanding technical requirements.

Please integrate the comment of reviewer #2 in the final version of the manuscript.

Kind regards,

Albert Rübben, Ass. Prof., M.D., Ph.D.

Academic Editor

PLOS ONE

Additional Editor Comments (optional):

The author have responded to the reviewer's comments and there remains only an additional minor change requested by the reviewer which should be corrected in the proofs.

Reviewers' comments:

Reviewer's Responses to Questions

**Comments to the Author**

1. If the authors have adequately addressed your comments raised in a previous round of review and you feel that this manuscript is now acceptable for publication, you may indicate that here to bypass the “Comments to the Author” section, enter your conflict of interest statement in the “Confidential to Editor” section, and submit your "Accept" recommendation.

Reviewer #2: (No Response)

2. Is the manuscript technically sound, and do the data support the conclusions?

Reviewer #2: Yes

3. Has the statistical analysis been performed appropriately and rigorously? 

Reviewer #2: Yes

4. Have the authors made all data underlying the findings in their manuscript fully available?

Reviewer #2: Yes

5. Is the manuscript presented in an intelligible fashion and written in standard English?

Reviewer #2: Yes

6. Review Comments to the Author

Reviewer #2: Comment:

Blood biomarkers are mentioned in line 117. I would suggest to mention which biomarkers were investigated also in text (Material and Methods) not only in table 2.

7. PLOS authors have the option to publish the peer review history of their article (what does this mean?). If published, this will include your full peer review and any attached files.

Reviewer #2: No

---

## [Editor Report · Acceptance letter]

27 Aug 2024

PONE-D-23-25220R1 

PLOS ONE

Dear Dr. Wikstrom, 

I'm pleased to inform you that your manuscript has been deemed suitable for publication in PLOS ONE. Congratulations! Your manuscript is now being handed over to our production team.

Kind regards, 

on behalf of

Albert Rübben 

Academic Editor

PLOS ONE